# High-Precision Mango Orchard Mapping Using a Deep Learning Pipeline Leveraging Object Detection and Segmentation

Muhammad Munir Afsar [1,*], Asim Dilawar Bakhshi [2], Muhammad Shahid Iqbal [3], Ejaz Hussain [1] and Javed Iqbal [1]

1   Institute of Geographical Information Systems, National University of Sciences and Technology, Islamabad 44000, Pakistan; ejaz@igis.nust.edu.pk (E.H.); javed@igis.nust.edu.pk (J.I.)
2   Department of Electrical Engineering, Military College of Signals, National University of Science and Technology, Rawalpindi 46000, Pakistan; asim.dilawar@mcs.edu.pk
3   Institute of Geo-Information and Earth Observation, ARID Agriculture University, Rawalpindi 46000, Pakistan; shahid.ucit@gmail.com
*   Correspondence: mafsar.phd18igis@igis.nust.edu.pk

**Abstract:** Precision agriculture-based orchard management relies heavily on the accurate delineation of tree canopies, especially for high-value crops like mangoes. Traditional GIS and remote sensing methods, such as Object-Based Imagery Analysis (OBIA), often face challenges due to overlapping canopies, complex tree structures, and varied light conditions. This study aims to enhance the accuracy of mango orchard mapping by developing a novel deep-learning approach that combines fine-tuned object detection and segmentation techniques. UAV imagery was collected over a 65-acre mango orchard in Multan, Pakistan, and processed into an RGB orthomosaic with a 3 cm ground sampling distance. The You Only Look Once (YOLOv7) framework was trained on an annotated dataset to detect individual mango trees. The resultant bounding boxes were used as prompts for the segment anything model (SAM) for precise delineation of canopy boundaries. Validation against ground truth data of 175 manually digitized trees showed a strong correlation ($R^2 = 0.97$), indicating high accuracy and minimal bias. The proposed method achieved a mean absolute percentage error (MAPE) of 4.94% and root mean square error (RMSE) of 80.23 sq ft against manually digitized tree canopies with an average size of 1290.14 sq ft. The proposed approach effectively addresses common issues such as inaccurate bounding boxes and over- or under-segmentation of tree canopies. The enhanced accuracy can substantially assist in various downstream tasks such as tree location mapping, canopy volume estimation, health monitoring, and crop yield estimation.

**Keywords:** precision agriculture; mango orchard mapping; canopy segmentation; UAV; object detection; YOLOv7; segment anything model

## 1. Introduction

Accurate delineation of individual tree canopies is critical for effective orchard management of high-value crops such as mangoes. Essential metrics like vegetation indices, tree crown area, and canopy volume depend on precise canopy measurements [1]. Imprecise and erroneous measurements can compromise the reliability and effectiveness of agricultural interventions. While field surveys provide the most accurate data, they are resource-intensive and have limited utility in an automation-driven precision agriculture paradigm.

Alternative strategies based on GIS and remote sensing facilitate end-to-end automation as well as precision and accuracy [2]. Schemes based on LiDAR data achieve canopy delineation by detecting tree tops and subsequently delineating individual tree crowns. However, due to the limited availability and high cost of LiDAR technology, there is an increased reliance on satellite and UAV imagery [3]. Manual delineation from aerial imagery, LiDAR, and stereo images is accurate for small areas but inefficient for large-scale assessments [4]. Even though LiDAR is a valuable data source for extracting tree crowns, it

constitutes a major expense for UAV platforms. Moreover, the quality of aerial imagery can be affected by weather and surface conditions.

Edge and region detection approaches enhanced with local maximum filtering and rule-based thresholding offer another alternative [5]. However, these methods are limited by variations in illumination and tree crown morphology. Watershed segmentation is noise-sensitive and may cause over-segmentation in dense forests [6]. Object-Based Image Analysis (OBIA) groups pixels into meaningful objects but is computationally intensive and requires manual intervention [4,7]. These techniques often struggle with overlapping canopies, complex tree structures, and varying light conditions [8].

Machine learning methods have improved segmentation accuracies, and state-of-the-art convolutional neural networks (CNNs) like YOLO, Mask R-CNN, fully convolutional networks (FCNs), and transformer-based models have contributed to better canopy segmentation [9–11]. Precise canopy size and structure provide early indications of tree stress, disease, or pruning effects [12,13]. It also facilitates targeted applications at the tree level, such as calibrating sprayers to minimize waste [14]. Accurate yield predictions are achieved by integrating tree crown area with canopy reflectance characteristics [15]. UAV-derived vegetation indices can predict yield at the individual tree level and accurate canopy delineation minimizes inaccuracies that might creep in due to undesired ground reflectance outside the canopy area while calculating tree health metrics [16].

Table 1 lists the most significant studies related to mango orchards using a diverse set of data and technologies. Despite these advances, challenges persist when using a single technique, especially for large trees like mangoes [17]. These primarily include overlapping canopies, and over- or under-segmentation [8,15]. The irregular morphology of tree species like mangoes leads to errors [18]. Therefore, diverse canopy structures and branching patterns require fully or semi-adaptable algorithms [5,9]. Occlusion also complicates segmentation, obscuring true crown boundaries and causing underestimations [19,20]. Spectral and textural properties in the background can lead to misclassifications [18]. Three-dimensional reconstruction techniques from UAV imagery in the case of dense canopies, such as structure-from-motion (SfM) photogrammetry, often struggle to process digital surface models accurately [3]. High-resolution UAV imagery sometimes results in undesired pruning, leading to sparse canopy centers [13].

A major challenge, however, is the availability of high-quality, labeled training data with enough variability, including occlusion, overlaps, and shadows [10,11]. Training these models requires significant resources, and their black-box nature can hinder refinement and trust [18,21]. YOLO and transformer-based models like Meta's SAM offer potential improvements to these challenges. YOLO performs well in tree detection with speed and fewer resources [6]. SAM's zero-shot and one-shot learning capabilities generalize well across diverse datasets, reducing reliance on extensively annotated datasets [19].

This study introduces a novel sequential approach for accurately delineating mango tree canopies in high-resolution UAV images. The speed of the YOLOv7 [22] object detection framework is combined with the precision of the segment anything model (SAM) [23] to locate trees and subsequently refine canopy boundaries quickly. Consequently, common issues such as inaccurate bounding boxes and over-segmentation are effectively addressed. A two-stage evaluation strategy is adopted to gauge the efficacy of the pipeline. The object detection stage is evaluated on a validation dataset split after the convergence of weights at the end of the transfer learning cycle. An optimal checkpoint is then used for bounding box regression. The final output is subsequently evaluated against a ground truth dataset comprising 175 mango trees. This comprehensive testing approach ensures that the proposed hybrid method delivers good precision and accuracy, and is ready to be integrated into diverse precision agriculture applications.

**Table 1.** Significant studies related to mango trees or orchards using diverse data sources and techniques.

| Research Topic | Technology | Strengths and Limitations |
|---|---|---|
| Acreage estimation of mango orchards [24] | EO-1 Hyperion hyperspectral data | Focuses on the characteristics of data source; lacks technical robustness. |
| Mango yield mapping at orchard scale [25] | UAV Photogrammetry, OBIA, Predictive Models | Accurate yield mapping and tree detection; manual tree crown delineation is time-consuming; variability in production estimation across cultivars. |
| Mango orchard age categorization [26] | Sentinel-2 data, OBIA, NDVI, ReNDVI, LSWI | Effective age categorization using vegetation indices; lower accuracy in classifying young orchards. |
| Tree species classification using ML [27] | LiDAR, Hyperspectral, Neural Networks, SVM | High accuracy with combined data; limited generalizability due to specific LiDAR metrics and data aggregation. |
| Orchard discrimination in Khairpur District, Pakistan [28] | Sentinel-1/2, Spectral Analysis, RF, SVM, Multi-Temporal Fusion | Discriminates between mango and date palm orchards; method not generalizable to canopies of other species |
| Object recognition in ecological environments [11] | PakSat-1R, NDVI, RetinaNet, CanopyNet, Multispectral | Accurate canopy detection in occlusion; no canopy delineation within the bounding box |
| Individual tree detection and counting [17] | High-Resolution Imagery, Canopy Height Model, Deep Learning | High accuracy in urban areas; no canopy delineation |
| Mapping horticultural tree structures [29] | ALS, TLS, Leaf Area Density, Vertical Leaf Area | ALS provided accurate crown structural parameters, but underperformed in crown volume estimation compared to TLS. |
| Height estimation of Mango and Avocado trees [30] | ALS, TLS, UAV-based RGB, Multi-spectral | UAV imagery provided height measurements comparable to TLS but ALS was more effective for large-area coverage. |
| Potential of WorldView-3 imagery for Mango yield estimation [15] | WorldView-3, ANN, TCA, NDVI | Outperformed traditional methods for satellite imagery. |

## 2. Methodology and Materials

The stage-wise workflow of the proposed methodology is shown in Figure 1. Modules for geospatial data collection, canopy detection and segmentation, and operationalization of downstream target problems are connected through preprocessing middleware.

### 2.1. Geospatial Module

The process begins with raw data collection using UAV flights, followed by processing through the software SiFT 1.2. While the software generates a variety of outputs, including multispectral reflectance mosaics, point clouds, digital surface, and terrain models, the RGB orthomosaic imagery is specifically utilized in the subsequent processing stage. This choice is driven by the requirement of the object detection and segmentation models employed, which necessitate the use of three-channel RGB images for optimal performance.

#### 2.1.1. Site Selection and Initial Setup

Multan is the prime mango-producing district of Pakistan. A 65-acre mango orchard under the management supervision of Mango Research Institute Multan (MRI), Pakistan was selected for the study, as shown in Figure 2 (left). The selected mango orchard exemplifies typical mango farming practices in the region with mango trees belonging to various cultivars and age groups. A total of 175 control trees representing the cultivars and age groups were selected and their positions, height, and canopy area were recorded through field survey.

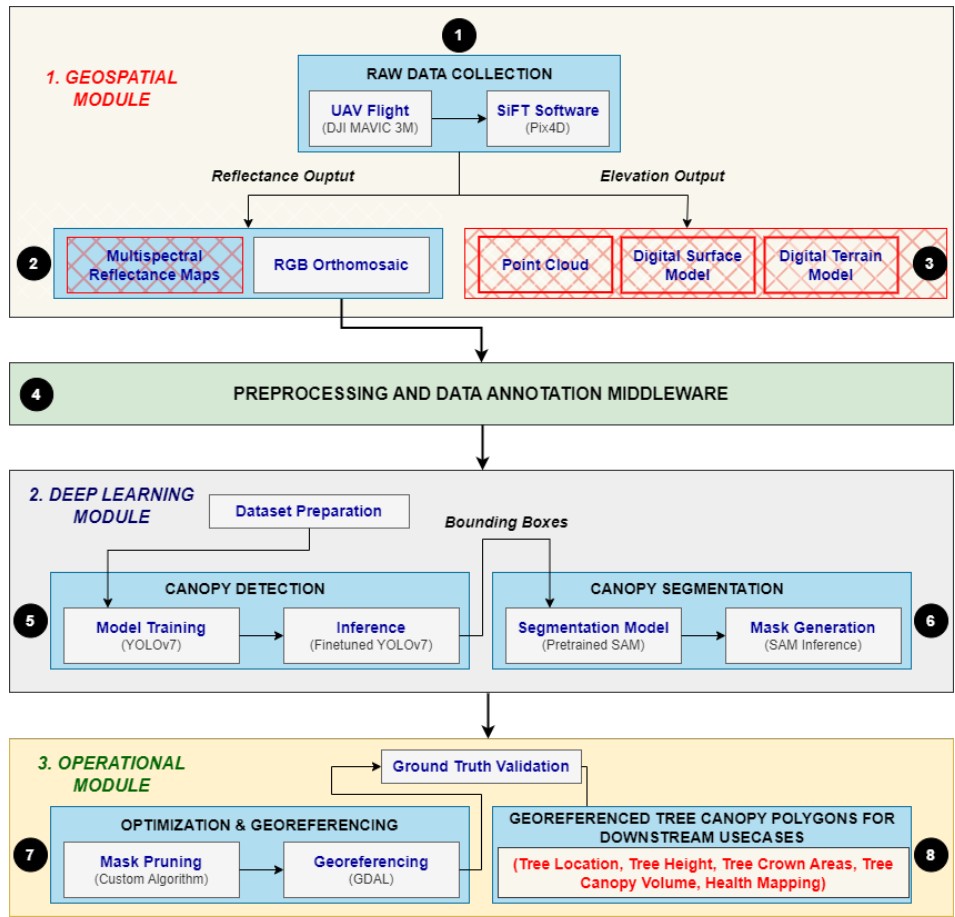

**Figure 1.** Workflow of the proposed methodology, including geospatial data collection, preprocessing, and deep learning for canopy detection and segmentation, followed by operational optimization and georeferencing.

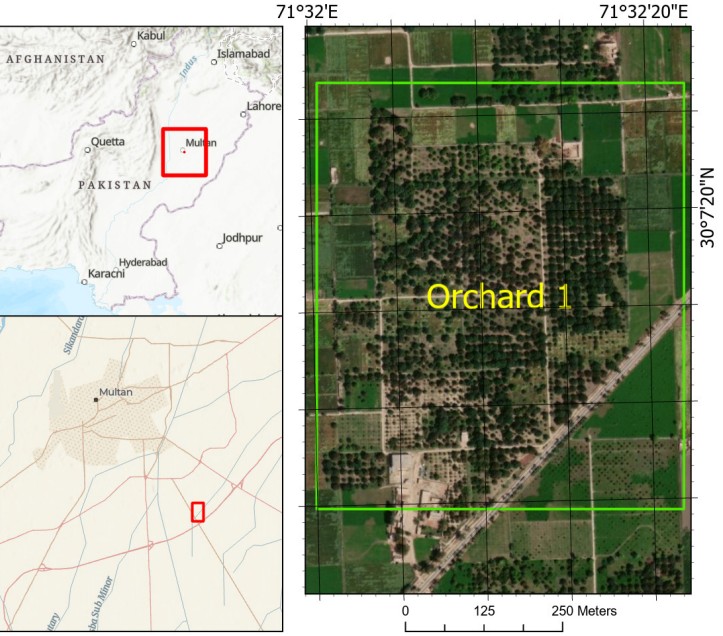

**Figure 2.** Geographical location and detailed view of the chosen orchard site near Multan, Pakistan; (**Left**): regional context, with the orchard's location marked in red boxes, (**Right**): high-resolution satellite image of Orchard 1 outlined by a green boundary.

### 2.1.2. Equipment and Software

- UAV and Realtime Kinematics (RTK) Mobile Station: The DJI Mavic 3M drone with its integrated advanced imaging and positioning technologies is an affordable platform for precision agriculture applications. It features a 4/3 CMOS RGB camera with 20 MP resolution, saving images in JPEG and RAW formats, and a multispectral camera with a 1/2.8-inch CMOS sensor, 5 MP resolution, capturing images in the Green (560 nm), Red (650 nm), Red Edge (730 nm), and NIR (860 nm) bands, saved in TIFF format. The sunlight sensor records solar irradiance for accurate post-processing of images. It is paired with the DJI D-RTK 2 Mobile Station, which supports all major GNSS systems and provides real-time differential corrections; the setup achieves 1 cm horizontal and 2 cm vertical positioning accuracy, crucial for precise mapping and data collection in the agricultural context. The drone uses its own flight planning and execution application to optimize flight parameters and patterns.
- Hardware and Software: Post-flight processing of UAV imagery requires good computer hardware. A system equipped with Intel(R) Core(TM) i9-10900KF CPU @ 3.70 GHz with 10 cores and 20 logical processors, 128 GB of RAM, 2 Terabyte Solid State Drive (SSD), and Nvidia RTX 3090 GPU was used. For processing the imagery, Pix4D 2.0 software, which utilizes Scale-Invariant Feature Transform (SIFT) algorithms was employed, ensuring high accuracy in image stitching and analysis. Other software includes ESRI's ArcGIS Pro 3.2, PyTorch 2.0, Torchvision 0.8, OpenCV 4.1.1, SciPy 1.7.0, and Cuda 11.8 Version 18 library.

### 2.1.3. Conduct of UAV Flights

A UAV flight over Orchard 1 was undertaken on 19 May 2024 with an altitude of 100m above ground level and an 80% forward and 70% side overlap (Figure 3). For enhanced positional accuracy, D-RTK 2 Mobile Station was used. The flight control software automatically computed the rest of the parameters. A total of 407 RGB JPEG images and 1628 multispectral TIFF images were taken. For this study, only RGB images have been processed into orthomosaic, with a ground sampling distance of 3 cm, and projected in UTM Zone 42N/WGS84. The resulting elevation products were not used for this study and only orthomosaic was utilized for canopy detection and segmentation.

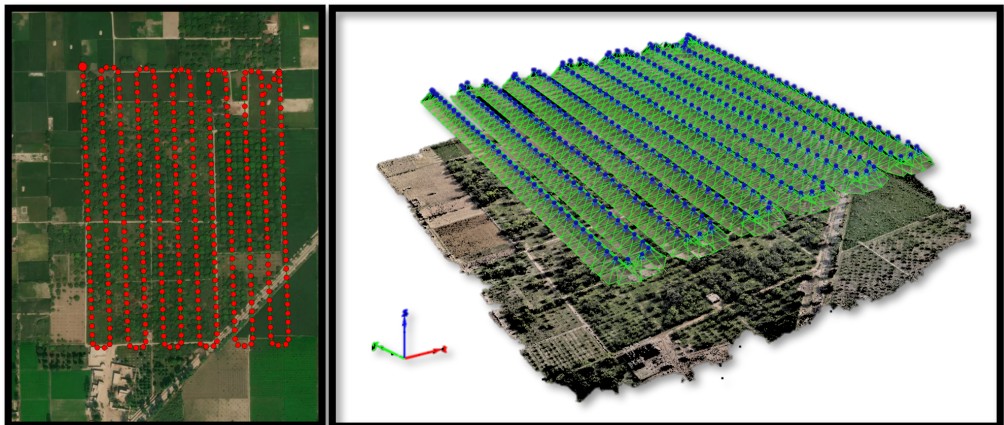

**Figure 3.** (**Left**): Flight path of DJI Mavic 3M. Every red dot presents where the UAV took an image. (**Right**): Perspective view during initial bundle adjustment and point cloud generation in Pix4D 2.0. Blue dots present the position where the UAV took an image, while green lines indicate the projected view of each photo on the ground.

### 2.1.4. Composite Image Model and Problem Formulation

Imagery bands $B$: blue, $G$: green, and $R$: red combine to generate a true color composite image. Each resultant pixel $(x, y)$ in the image can be represented as:

$$I(x, y) = [R(x, y), G(x, y), B(x, y)]$$

Given $I(x, y)$ as the working image model, the target problems for the next stage are defined as:

- Object Recognition: The objective is to predict a bounding box around each mango canopy. For each detected canopy, a bounding box $BB = (x_{\min}, y_{\min}, x_{\max}, y_{\max})$ is to be automatically inferred, where $(x_{\min}, y_{\min})$ and $(x_{\max}, y_{\max})$ are the coordinates of the top-left and bottom-right corners of the bounding box, respectively.
- Segmentation: For a given bounding box $BB$, the segmentation problem involves predicting a binary mask $S(x, y)$ such that:

$$S(x, y) = \begin{cases} 1 & \text{if } (x, y) \text{ is within the mango canopy edge boundary} \\ 0 & \text{otherwise} \end{cases} \tag{1}$$

### 2.2. Preprocessing and Data Annotation Middleware

This intermediate stage is crucial for staging the data for the deep-learning pipeline.

### 2.2.1. Image Annotation

Composite imagery is annotated to create a labeled training dataset for object detection, as shown in Figure 4. This was performed using the visual geometry group image annotator (VIA), an open-source tool to define image regions. The annotated chips are then converted into ground truth files using custom Python scripts to ensure consistency and accuracy.

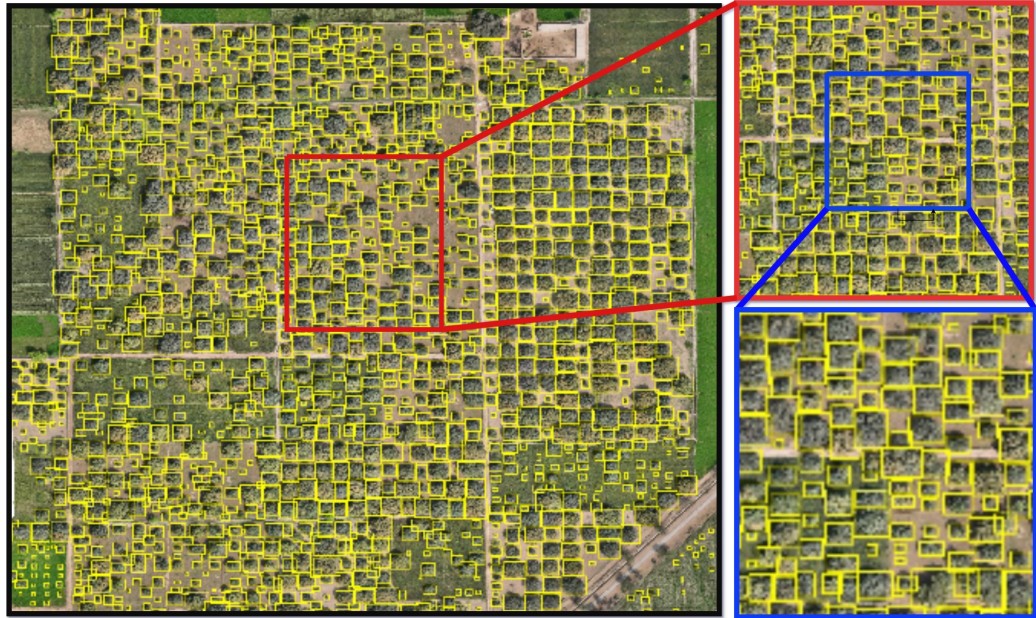

**Figure 4.** Composite image annotation aimed to enclose each of the 1871 mango tree canopies in a bounding box.

Given an image $I(x, y)$, the annotation process results in a set of bounding boxes $\{BB_i\}$ for each mango tree, where each $BB_i = (x^i_{\min}, y^i_{\min}, x^i_{\max}, y^i_{\max})$ represents the coordinates of the top-left and bottom-right corners of the bounding box.

2.2.2. Label Transformation

Obtained bounding box coordinates need to be properly transformed such that for a particular bounding box $BB_i$, the conversion is given by:

$$(x^i_{\text{center}}, y^i_{\text{center}}, w_i, h_i) = \left( \frac{x^i_{\min} + x^i_{\max}}{2}, \frac{y^i_{\min} + y^i_{\max}}{2}, x^i_{\max} - x^i_{\min}, y^i_{\max} - y^i_{\min} \right) \quad (2)$$

2.2.3. Image Tiling, Dataset Composition, and Merging

The orthomosaic image used in this study was exported to PNG format and divided into 16 tiles, each with dimensions of $800 \times 800$ pixels. This tiling was necessary for two primary reasons: (1) to ensure that each tree had sufficient visual detail for training the object detection model, and (2) to manage GPU VRAM limitations during processing. Table 2 provides a summary of how the tiles were allocated between training, validation, and test splits.

**Table 2.** Dataset composition for model training, validation, and testing.

| Dataset | Number of Tiles |
|---|---|
| Training Split | 14 |
| Validation/Test Splits | 2 |
| Total Tiles | 16 |

This ensured that 87.5% of individual trees (1637) were included in the training sample and 12.5% (234) of trees were kept for validation of the output. Detections were performed on each tile measuring $800 \times 800$ pixels. YOLOv7 was used to detect individual mango trees within each tile. The results from all tiles were then appended together to generate a complete detection map for the full orthomosaic image. However, it is important to note that for medium- to low-resolution satellite data and/or detecting more than one land use class, a larger number of images/image tiles will be required to train a Yolo7 variant.

After the detection process, the complete orthomosaic image was segmented using the segment anything model (SAM). The segmentation process generated masks for all potential tree canopies within the image.

The segmentation masks were compared against the detection bounding boxes from YOLOv7. Masks whose centers did not fall within any detection bounding box were discarded, ensuring that only relevant tree canopies identified by the object detection model were retained. This process allowed for the integration of the detection and segmentation results into a comprehensive mango tree detection map covering the entire orchard.

*2.3. Deep Learning Module*

This module comprises two sequential stages: (i) canopy detection and (ii) segmentation. The canopy detection stage involves dataset preparation, followed by a custom object detection model training and then inference with this fine-tuned model to generate $BB_i$ for individual mango trees. The intermediate model with the best validation scores was selected as the final model from the training cycle. In the subsequent canopy segmentation stage, a pre-trained object segmentation model is used for mask generation for each tree $S_i(x, y)$.

2.3.1. Canopy Detection

The canopy detection process involves training a YOLO variant to detect mango tree canopies from aerial images. Unlike traditional object detection methods that apply a sliding window or region proposal network to identify objects, YOLO formulates a single regression problem, directly predicting bounding boxes and class probabilities from full images in one evaluation. This unified architecture allows YOLO to process images rapidly, making it highly efficient for real-time applications as well. The model divides

an image into a grid and predicts bounding boxes and probabilities for each grid cell, leveraging convolutional neural networks (CNNs) to extract spatial features and contextual information. YOLO's capability to balance precision and computational efficiency makes it an ideal choice for tasks requiring high-speed object detection across various domains.

The variant YOLOv7, chosen for its balance between speed and accuracy, is utilized in this process. The training process aims to optimize the model by minimizing the loss function $\mathcal{L}$, which is generally a combination of classification, localization, and confidence losses:

$$\mathcal{L} = \mathcal{L}_1 + \mathcal{L}_2 + \mathcal{L}_3 \tag{3}$$

where $\mathcal{L}_1$ is the classification loss, $\mathcal{L}_2$ is the localization loss calculated as the sum of squared errors between the predicted and ground truth bounding box coordinates, and $\mathcal{L}_3$ is the confidence loss regarding presence of an object within a predicted bounding box.

During training, the model's parameters are updated using backpropagation and stochastic gradient descent (SGD). The trained YOLOv7 model predicts a bounding box $BB_i$ for each detected canopy with associated confidence scores. Non-maximum suppression (NMS) is used to handle redundant boxes and retain the most confident predictions.

### 2.3.2. Canopy Segmentation

After detecting the bounding boxes, the next step is to generate precise masks for each mango canopy using the SAM. It is a fairly recent object segmentation framework designed to provide precise boundary delineation for objects within an image. Rather than extensive fine-tuning on specific datasets, SAM employs a zero-shot learning approach, enabling it to generalize well across diverse image domains without additional training. SAM uses a combination of CNNs and transformer-based architectures to capture detailed contextual and spatial information. In agricultural monitoring where precise segmentation is critical, SAM [23] adequately identifies fine-grained edges.

The integration of SAM and YOLO significantly reduces the false positive rate and provides accurate segmentation boundaries for the mango canopies. The detections from the YOLOv7 model are used to refine the SAM detections. Any segmentation mask from SAM that does not have significant overlap with a YOLO detection is discarded. This process helps reject false positives from background regions and tree shadows, improving the accuracy of the final results.

For each detected bounding box $BB_i$, a cropped image patch $I_i$ is extracted from the original image $I(x, y)$. This patch $I_i$ is then fed into the SAM model to generate a binary mask $S_i(x, y)$, as per (1).

### 2.3.3. Significance of the Deep Learning Pipeline

The pipeline leverages the strengths of both YOLO and SAM models to achieve high-precision mango canopy detection and segmentation. YOLO is employed first for its robust object detection capabilities, generating bounding boxes that accurately identify mango canopies. Fine-tuned for the specific task, this detection model effectively reduces background noise and eliminates false positives associated with non-canopy elements such as shadows and ground structures.

Following the bounding box detection, SAM performs the detailed segmentation of the mango canopies. SAM's ability to deliver precise boundary delineation is crucial for accurate health metric calculations. By focusing on the regions defined by YOLO, SAM can generate segmentation masks that are not only accurate but also free from extraneous background elements.

This sequential approach ensures that the initial detection phase effectively isolates the relevant objects, while the subsequent segmentation phase refines the boundaries with high precision.

*2.4. Operational Module*

This module addresses optimization and georeferencing to facilitate various downstream applications critical to precision agriculture. Mask pruning is performed using a custom algorithm, and georeferencing is accomplished using the GDAL [31] library. The processed data are then staged for several key tasks, including tree location identification, canopy area and volume estimation, and health mapping using vegetation indices.

### 2.4.1. Georeferencing

The pixel values of the segmented canopies are converted into georeferenced polygons using the GDAL Python library. This conversion allows the integration of canopy data with geographic information systems (GIS) for spatial analysis. Each pixel's position in the image is mapped to real-world coordinates, enabling precise location-based assessments of tree canopies across the orchard.

### 2.4.2. Canopy Area Estimation

The area of each tree canopy is defined as a sum of pixel areas within each georeferenced polygon. If each pixel represents a known ground area (e.g., $A_{\text{pixel}}$ in square meters), the canopy area ($A_{\text{canopy}}$) can be estimated as:

$$A_{\text{canopy}} = \sum_{i=1}^{n} A_{\text{pixel}_i} \tag{4}$$

where $n$ is the number of pixels within the canopy boundary.

### 2.4.3. Canopy Volume Estimation

Canopy volume is estimated by integrating the height of the canopy across its area. Given the digital surface model (DSM), the volume ($V_{\text{canopy}}$) can be estimated as

$$V_{\text{canopy}} = \sum_{i=1}^{n} A_{\text{pixel}_i} \times h_{\text{pixel}_i} \tag{5}$$

where $h_{\text{pixel}_i}$ is the height of the canopy at pixel $i$, derived from the DSM.

### 2.4.4. Health Mapping Using Vegetation Indices

Health mapping is modeled using two separate vegetation indices: the normalized difference vegetation index (NDVI) and the normalized difference red edge (NDRE) index. The NDVI is computed as:

$$\text{NDVI} = \frac{\text{NIR} - \text{R}}{\text{NIR} + \text{R}} \tag{6}$$

where NIR is the near-infrared band and R is the red band of the imagery.

NDVI is a good indicator of the general vigor of vegetation, with values getting lower as the vegetation is exposed to stress or disease. However, in later growth stages, NDVI may become less sensitive.

The NDRE is another important index, particularly useful for assessing crop health, especially in later growth stages where NDVI may become less sensitive. In such cases, we can resort to a measure of nutrient deficiency, for which NDRE is a good choice; it may be calculated as

$$\text{NDRE} = \frac{\text{NIR} - \text{RE}}{\text{NIR} + \text{RE}} \tag{7}$$

where RE represents the red edge band of the imagery which is sensitive to changes in chlorophyll content.

### 2.4.5. Accuracy Assessment

The accuracy of the segmentation results is validated against ground truth data. This involves comparing the detected canopy polygons with manually delineated ground truth polygons. The assessment is quantified using metrics such as Intersection over Union (IoU), precision, recall, and F1-score. GIS tools are employed to plot and visualize both the detection and segmentation results, providing a clear overview of the method's performance.

### 2.5. Evaluation Metrics

Several evaluation metrics were used to assess the performance of the proposed scheme and quantify the accuracy and robustness as well as the reliability of detection and segmentation.

### 2.5.1. Precision, Recall, and F1 Score

Precision is defined as the ratio of correctly identified tree canopies to the total number of claimed detection outcomes.

$$\text{Precision} = \frac{\text{True Positives}}{\text{True Positives} + \text{False Positives}} \tag{8}$$

Recall is the ratio of correctly identified tree canopies to the total number of actual positives. This is a realistic measure of the model's ability to correctly identify all relevant instances.

$$\text{Recall} = \frac{\text{True Positives}}{\text{True Positives} + \text{False Negatives}} \tag{9}$$

Being the harmonic mean of Precision and Recall, the F1 score provides a single trade-off metric, especially in the case of a problem like canopy detection, where the class distribution is imbalanced.

$$\text{F1 Score} = 2 \times \frac{\text{Precision} \times \text{Recall}}{\text{Precision} + \text{Recall}} \tag{10}$$

### 2.5.2. Mean Average Precision (mAP)

To evaluate the object detection precision, two variants of mAP are used. $\text{mAP}_{0.5}$ considers detections with IoU thresholds of 0.5, which effectively means a 50% overlap between predicted and ground truth bounding boxes. Another variant, $\text{mAP}_{0.5:0.95}$, averages the mAP over IoU thresholds, ranging from 0.5 to 0.95 in steps of 0.05. This is to provide a more comprehensive evaluation of model performance across varying levels of overlap.

### 2.5.3. Root Mean Square Error (RMSE)

The standard deviation of the prediction errors is computed as

$$\text{RMSE} = \sqrt{\frac{1}{n} \sum_{i=1}^{n} (\hat{y}_i - y_i)^2} \tag{11}$$

where $\hat{y}_i$, $y_i$, and $n$ are the predicted value, actual value, and total number of observations, respectively.

### 2.5.4. Bland–Altman Analysis

It is necessary to quantify the agreement between the results of the proposed scheme and the manual detections on the ground. The Bland–Altman analysis utilized the bias and the Limits of Agreement (LoA). For our study, the standard definition of LoA is defined as the mean difference $\pm$ 1.96 times the standard deviation of the differences. This is a reasonable range within which most differences between the methods are expected to lie.

2.5.5. Mean Absolute Percentage Error (MAPE)

The average absolute percentage difference between the predicted and actual values is given as:

$$\text{MAPE} = \frac{1}{n} \sum_{i=1}^{n} \left| \frac{y_i - \hat{y}_i}{y_i} \right| \times 100\% \tag{12}$$

where $y_i$, $\hat{y}_i$, and $n$ represent the manual measurement, the automated measurement, and the total number of observations, respectively.

By employing these metrics, the performance of the deep learning pipeline is comprehensively evaluated, ensuring that the model not only performs well on average but also maintains consistency and reliability across different scenarios.

## 3. Results and Discussion

### 3.1. Stage 1: Canopy Detection and Localization

3.1.1. Training the Object Detection Model

The YOLOv7 model [22] employed for tree canopy detection was trained using a meticulously chosen set of hyperparameters to optimize performance. Table 3 lists the essential hyperparameters utilized in the training regimen. Extensive experimentation and domain-specific considerations guided the selection of these hyperparameters. This approach ensured robust and efficient training of the YOLOv7 model for tree canopy detection. GPU acceleration was employed and the model underwent training for 1000 epochs, utilizing a batch size of 2 and an image resolution of $800 \times 800$ pixels. The learning rate, momentum, and weight decay were systematically adjusted to optimize the model's convergence and generalization performance.

**Table 3.** Essential hyperparameters for YOLOv7 model training.

| Hyperparameter | Value |
|---|---|
| Initial Learning Rate (*lr0*) | 0.01 |
| Final Learning Rate (*lrf*) | 0.1 |
| Momentum (*momentum*) | 0.937 |
| Weight Decay (*weight_decay*) | 0.0005 |
| Warmup Epochs (*warmupe_pochs*) | 3.0 |
| Warmup Momentum (*warmup_momentum*) | 0.8 |
| Warmup Bias Learning Rate (*warmup_bias_lr*) | 0.1 |
| Batch Size (*batch_size*) | 2 |
| Image Size (*img_size*) | [800, 800] |
| Number of Epochs (*epochs*) | 1000 |
| Device (*device*) | 0 |
| CPU Cores (*workers*) | 8 |

3.1.2. Performance Evaluation of Canopy Detection Model

The training cycle of the YOLO model reveals several key insights (Figure 5). The consistent decrease in both training loss (from 0.12 to around 0.04) and validation loss (from 0.12 to approximately 0.08) over the epochs indicates that the model is effectively learning and optimizing its parameters. The precision, recall, and F1 score trends initially exhibit variability but stabilize around 0.8, 0.75, and 0.77, respectively, after 400 epochs. This stabilization reflects the model's improved accuracy and completeness in detecting mango tree canopies, indicating a balanced trade-off between precision and recall, which is essential for accurate canopy localization.

The $\text{mAP}_{0.5}$ and $\text{mAP}_{0.5:0.95}$ metrics demonstrate significant improvement in detection performance across varying IoU thresholds. The $\text{mAP}_{0.5}$ metric rises sharply to about 0.8, indicating that the model can accurately detect mango canopies with at least 50% overlap. The $\text{mAP}_{0.5:0.95}$ metric, which stabilizes around 0.35, shows the model's performance across stricter overlap criteria, highlighting its robustness and reliability. Finally, the

accuracy trend, calculated as the average of precision and recall, increases from 0.1 to 0.8, demonstrating a substantial improvement and stabilization in the model's localization capabilities over the training period.

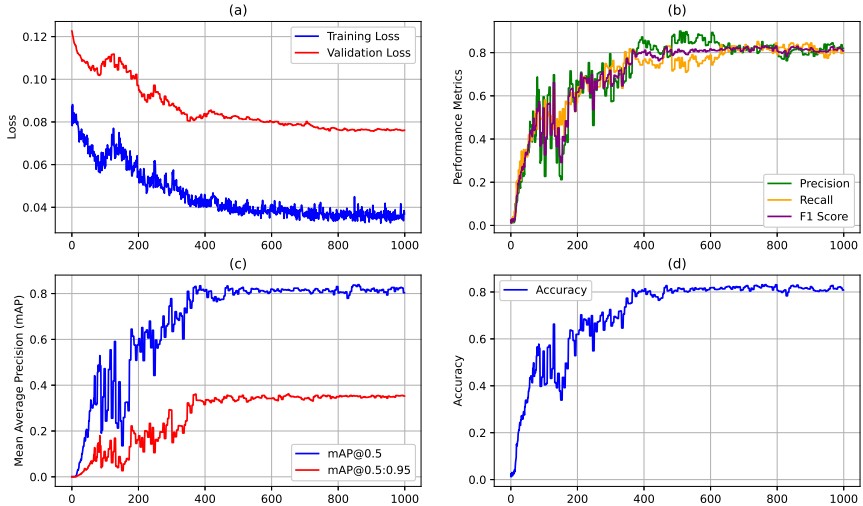

**Figure 5.** Overview of the YOLOv7 model training process for mango tree canopy detection. (**a**) Training and validation loss trends, showing the convergence and generalization capabilities, (**b**) precision, recall, and F1 score trends over the training cycle to reflect, (**c**) Mean Average Precision (mAP) at IoU thresholds of 0.5 and 0.5:0.95, indicating the model's detection performance across varying overlap levels, and (**d**) accuracy trend, calculated as the average of precision and recall, demonstrating the overall performance improvement of the model during training.

### 3.1.3. Triadic Checkpoint Selection Criteria for Deep Learning Pipeline

Since the final segmentation of canopies and consequently the application endpoint is highly dependent on the initial canopy detection state, it is not straightforward to use the final trained model. To address this problem, triadic criteria $\zeta$ were set for the model checkpoint finalization, i.e., a balance between overfitting minimization, the precision of bounding box regression, and robustness. Lowest validation loss ($\zeta_1$), highest $\text{mAP}_{0.5}$ ($\zeta_2$), and highest $\text{mAP}_{0.5:0.95}$ ($\zeta_3$) were used as parameters to represent these three criteria, respectively.

Table 4 lists the evaluation of model checkpoints based on each criterion. The checkpoint with $\min(\zeta_1) = 0.07571$ was observed at epoch 905, with $\zeta_2 = 0.8235$ and $\zeta_3 = 0.3524$. This checkpoint indicates excellent model performance in terms of minimizing overfitting and maintaining good generalization.

**Table 4.** Triadic checkpoint selection criteria based on validation loss ($\zeta_1$), $\text{mAP}_{0.5}$ ($\zeta_2$), and $\text{mAP}_{0.5:0.95}$ ($\zeta_3$).

| Criterion | Epoch | $\zeta_1$ | $\zeta_2$ | $\zeta_3$ |
|---|---|---|---|---|
| $\min(\zeta_1)$ | 905 | 0.07571 | 0.8235 | 0.3524 |
| $\max(\zeta_2)$ | 859 | 0.07631 | 0.8388 | 0.3504 |
| $\max(\zeta_3)$ | 649 | 0.07920 | 0.8176 | 0.3618 |

For the $\max(\zeta_2) = 0.8388$, the best checkpoint was at epoch 859. Hence, this variant was most accurate in detecting mango canopies with at least 50% overlap, and despite a slightly higher validation loss compared to the lowest validation loss checkpoint, this epoch achieved the highest precision in canopy detection.

The checkpoint with $\max(\zeta_3) = 0.3618$ was found at epoch 649. This checkpoint reflects the model's robust performance across a range of IoU thresholds, indicating its versatility and reliability in varying detection scenarios.

The final choice depends on the specific requirements of the downstream use case. If minimizing error and overfitting is the priority, the checkpoint at epoch 905 is ideal. However, if achieving the highest detection accuracy is more critical, the checkpoint at epoch 859 would be preferable. For agriculture applications requiring balanced performance across different overlap criteria, the checkpoint at epoch 649 is the best choice.

For this study, the checkpoint at epoch 905 was used for further experiments at the canopy segmentation stage.

### 3.2. Stage 2: Canopy Segmentation

The segment anything model (SAM) adequately handles multi-modal input, including images with bounding boxes or key point data. In this stage, the output of the canopy detection process from YOLO is utilized as a multi-modal prompt for SAM, allowing for detailed and precise segmentation.

To optimize performance, the large version of the visual transformer within SAM was selected. This choice was informed by preliminary evaluations which indicated that the base model, while less computationally demanding, did not provide the requisite accuracy for our specific application. The detailed hyperparameters used for SAM inference are enumerated in Table 5.

**Table 5.** Parameter settings for the SAM model for inferring canopy segmentation masks from bounding box prompts.

| Parameter | Value |
| --- | --- |
| points_per_side | 32 |
| points_per_batch | 64 |
| pred_iou_thresh | 0.88 |
| stability_score_thresh | 0.95 |
| stability_score_offset | 1 |
| box_nms_thresh | 0.7 |
| crop_n_layers | 0 |
| crop_nms_thresh | 0.7 |
| crop_overlap_ratio | 512 / 1500 |
| crop_n_points_downscale_factor | 1 |

These hyperparameters are pivotal for refining the canopy segmentation performance. The `points_per_side` parameter determines the number of points sampled along each side of the bounding box. This significantly influences the resolution of the canopy boundaries, enabling precise tracing of tree edges. The parameter `points_per_batch` sets the number of points processed in each batch, ensuring that the model handles large-scale canopy bounding boxes effectively. The `pred_iou_thresh` parameter specifies the IoU threshold, ensuring high precision in generating segmentation masks by accurately distinguishing tree canopies from the background. The `stability_score_thresh` and `stability_score_offset` parameters are crucial for segmentation mask sensitivity fine-tuning. The `box_nms_thresh` controls the non-maximum suppression threshold for bounding boxes, eliminating redundant detections and retaining the most accurate canopy boundaries. Parameters such as `crop_n_layers`, `crop_nms_thresh`, and `crop_overlap_ratio` allow effective capturing of multiple scales and overlapping regions to handle ecological occlusion. Lastly, the `crop_n_points_downscale_factor` reduces the number of points within each crop, managing computational complexity while maintaining segmentation accuracy.

SAM enhances the delineation of enclosed regions within the provided bounding boxes, transforming them into precise segmentation masks that meticulously trace the edges of each mango tree canopy (Figure 6). This refinement significantly improves the

differentiation of individual canopies, facilitating the analysis of specific tree canopy characteristics such as shape, size, or area.

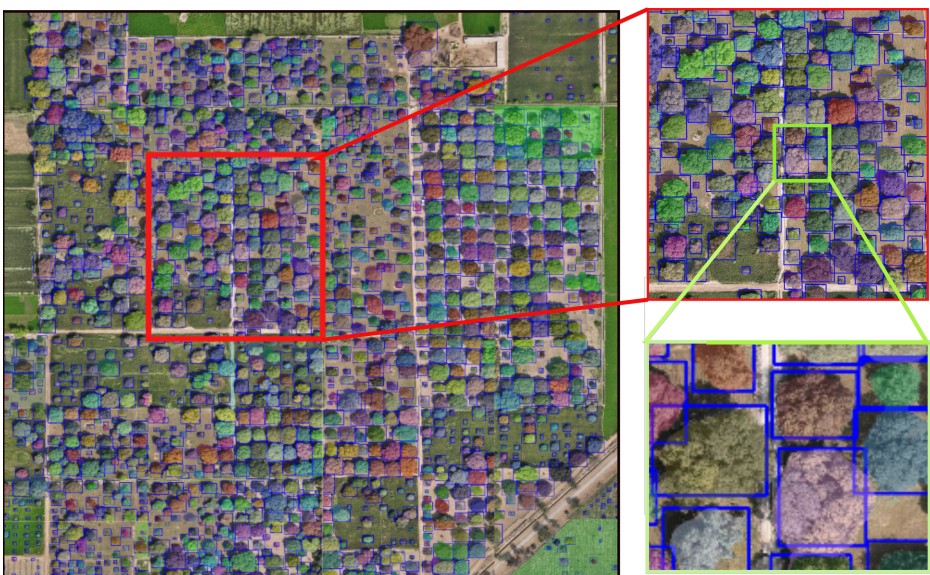

**Figure 6.** Segmentation output of SAM; each segmented tree is shown with a different-colored segmentation mask.

*3.3. Validation Over Field Survey Dataset*

3.3.1. Ground Truth Dataset

To validate the effectiveness of the hybrid approach, the segmented canopies were compared against ground truth data collected from field surveys. A study was conducted on 175 mango trees in another mango orchard, whose canopies were manually digitized (Figure 7) from orthomosaic images using ArcGIS Pro 3.2, and their areas $A_{\mathrm{manual}}$ were calculated in square feet. The corresponding composite imagery of the orchard section was fed to the proposed deep learning pipeline. $A_{\mathrm{manual}}$ was then compared to the output of the pipeline, i.e., automatically delineated canopy areas $A_{\mathrm{auto}}$.

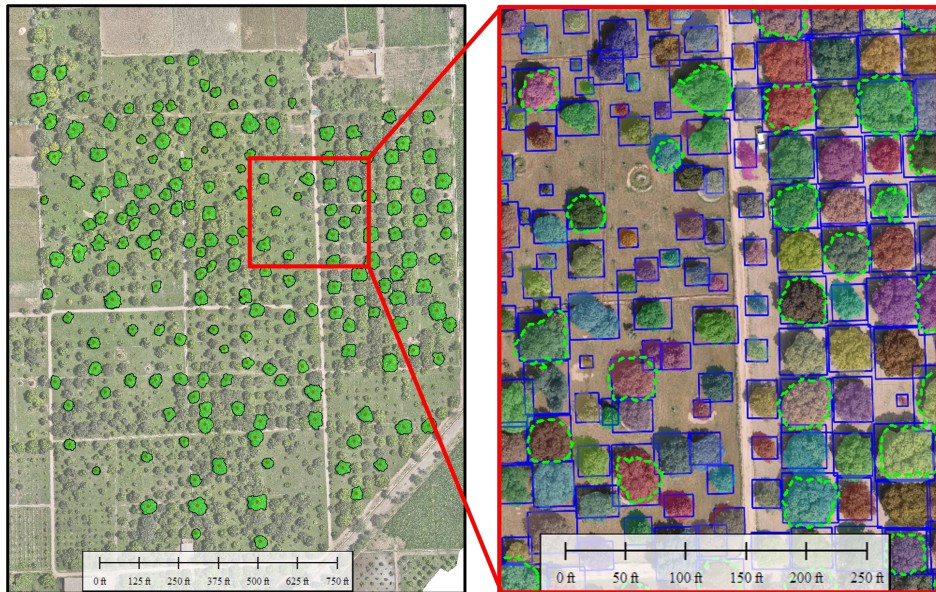

**Figure 7.** Green areas are manually digitized canopies and red dots are the center points of the canopy mass; a patch is zoomed in on to show how the manually traced edges are validated against the output of pre-trained SAM.

### 3.3.2. Performance Validation

Figure 8 shows the validation results of the proposed deep learning pipeline against the ground truth dataset using four different metrics.

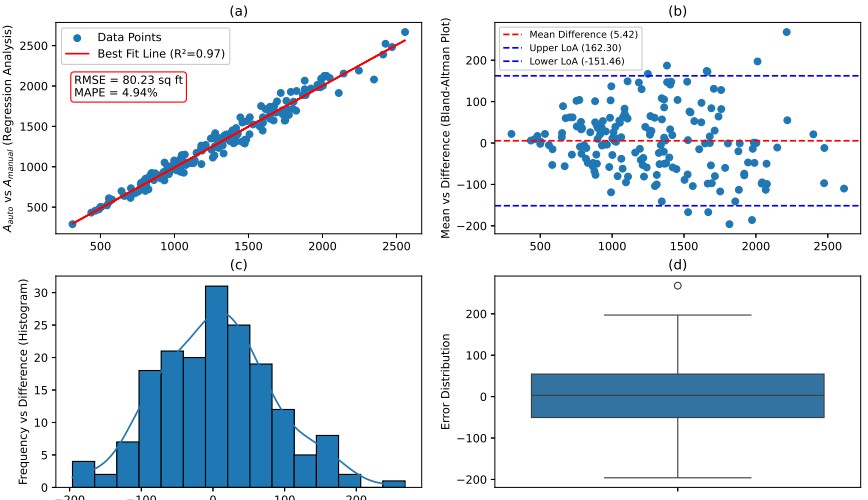

**Figure 8.** A comprehensive validation of tree canopy detection and segmentation results using the proposed automated pipeline. (**a**) Regression Analysis: Scatter plot depicting the relationship between manually digitized canopy areas ($A_{\mathrm{manual}}$) and automatically delineated canopy areas ($A_{\mathrm{auto}}$). (**b**) Bland Altman Plot: Scatter plot showing the mean of $A_{\mathrm{manual}}$ and $A_{\mathrm{auto}}$ on the x-axis and the difference ($A_{\mathrm{manual}} - A_{\mathrm{auto}}$) on the y-axis. (**c**) Histogram of Differences: Histogram illustrating the distribution of differences between $A_{\mathrm{manual}}$ and $A_{\mathrm{auto}}$. (**d**) Error Distribution (Box Plot): Box plot summarizing the errors between $A_{\mathrm{manual}}$ and $A_{\mathrm{auto}}$.

The regression analysis demonstrates a strong linear correlation between manually digitized ($A_{\mathrm{manual}}$) and automatically delineated canopy areas ($A_{\mathrm{auto}}$). The best-fit line, with an $R^2 = 0.97$, indicates that 97% of the variance in the automated measurements can be explained by the manual measurements with minimal deviation. The RMSE of 80.23 sq ft against the average canopy size of 1290.14 sq ft computed over the complete ground truth dataset suggests decent efficacy of the proposed scheme.

For Bland–Altman analysis, the mean difference is 5.42 sq ft, suggesting an insignificant systematic bias in the automated measurements. The limits of agreement (LoA), calculated as the mean difference $\pm 1.96$ times the standard deviation, provide a range of $\pm 155.23$ sq ft, within which most differences between the manual and automated measurements lie. The majority of data points fall within these limits, confirming consistent agreement across the range of measurements. However, there are a few outliers that indicate discrepancies for certain tree canopies, which warrant further investigation.

The histogram of differences confirms that most differences cluster around zero, reinforcing the minimal bias observed earlier. Errors are normally distributed, as desirable in validating the reliability of the pipeline. However, the tails of the distribution indicate some instances of larger discrepancies, which might be due to variations in tree shapes or shadows affecting the automated detection.

The median difference of errors is close to zero, and the interquartile range is 53.00 sq ft, indicating that the majority of errors are small. The presence of a few outliers again suggests occasional significant discrepancies. These outliers could be due to specific characteristics of certain tree canopies, such as overlapping branches or irregular shapes that challenge the automated segmentation process.

Lastly, the mean absolute percentage error (MAPE) was computed using (11) to provide unified metrics for assessment such as $y_i = A_{\mathrm{manual},i}$ and $\hat{y}_i = A_{\mathrm{auto}}$, and $i = 0, 1, \cdots, 174$ denote the tree canopies in the ground truth dataset. The MAPE of 4.8%

demonstrates that the automated method achieves a high level of accuracy, closely aligning with the manually measured canopy areas. However, a relatively high RMSE of 80.23 square feet also points to significant variability in the errors. Overall, both results in tandem suggest that while the automated segmentation method performs well on average, it does exhibit some anomalous measurement errors, indicating that further refinement is necessary according to the dictates of the target application.

### 3.3.3. Potential Limitations of the Pipeline

While the results are promising, there are some potential limitations. The presence of outliers suggests that the automated pipeline occasionally struggles with certain tree canopies. This might be due to specific challenges like overlapping branches or non-standard shapes. Moreover, while the dataset of 175 trees provides a reasonable sample, further validation with a larger and more diverse dataset would certainly strengthen the reliability of the results. Lastly, variations in lighting, shadowing, and image resolution could affect the accuracy of the automated measurements. This necessitates the introduction of a novel and robust preprocessing sub-module to mitigate these effects.

### 3.4. Potential Applications of the Proposed Framework

To illustrate the broader potential of the proposed scheme, Figure 9 presents the target applications of tree height mapping and canopy volume estimation using DSM based on photogrammetric 3D point cloud. Tree canopies are color-coded according to height in Figure 9a. The gradient shifts from green to red as the height increases. Figure 9b shows the canopy height in meters, whereas the highest points are marked in Figure 9c. This shows the potential for a precise understanding of the canopy structures. Lastly, Figure 9d gives an estimate of the canopy volume using a DSM. The red indicates lower volume and increases towards green. Such volumetric analysis can aid in assessing biomass, which can subsequently correlate with yield estimation.

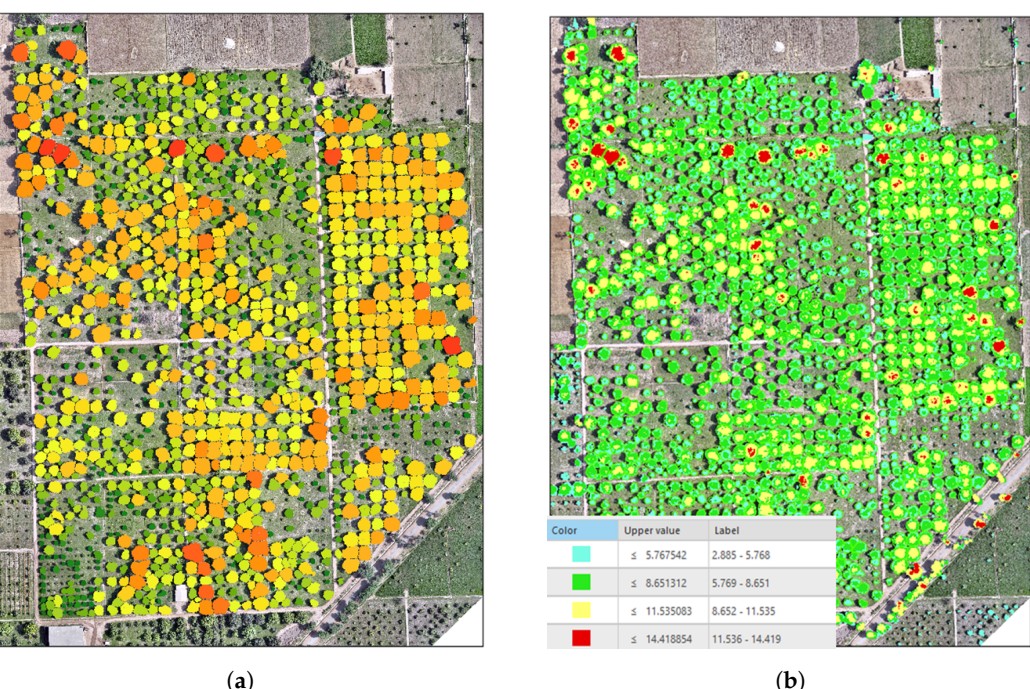

(**a**)                                                              (**b**)

**Figure 9.** *Cont.*

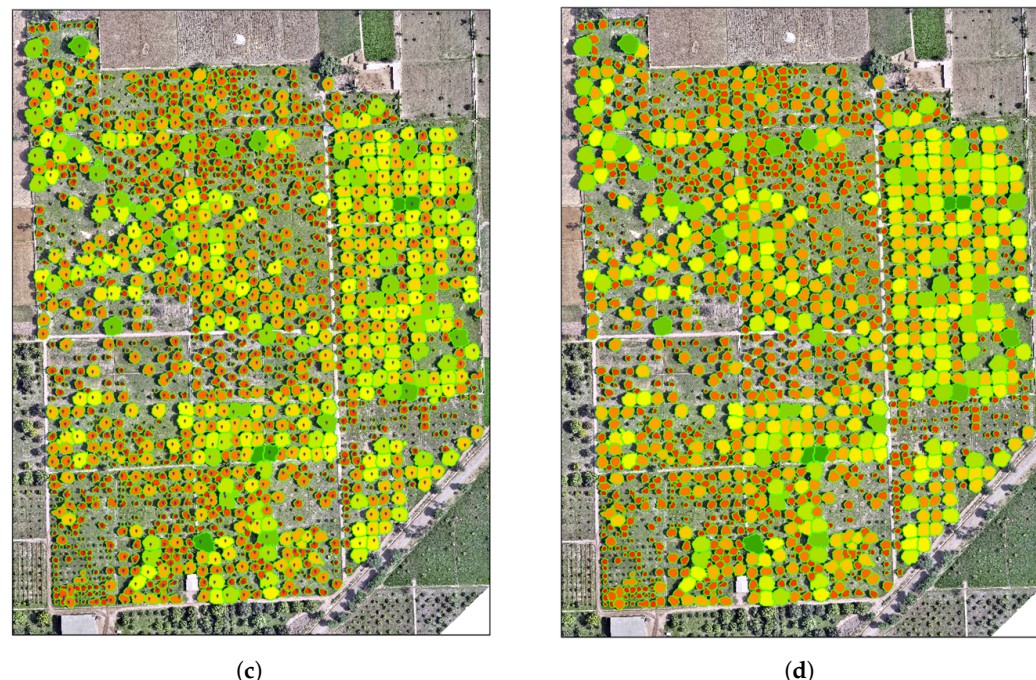

(**c**)          (**d**)

**Figure 9.** Canopy height and volume derived from DSM based on photogrammetric 3D point cloud and accurate canopy delineation through the proposed framework. (**a**) Height of canopies (green to red). (**b**) Height of canopy (in meters). (**c**) The highest point of the canopy. (**d**) Canopy volume using DSM (red to green).

Figure 10a shows how the NDVI can be used for the health of mango trees. The brown canopy gradient indicates weaker trees, while the green canopy gradient suggests stronger and healthier trees. This gives ample opportunities for monitoring the overall health of the orchard. The NDRE provides further insights regarding health in Figure 10b.

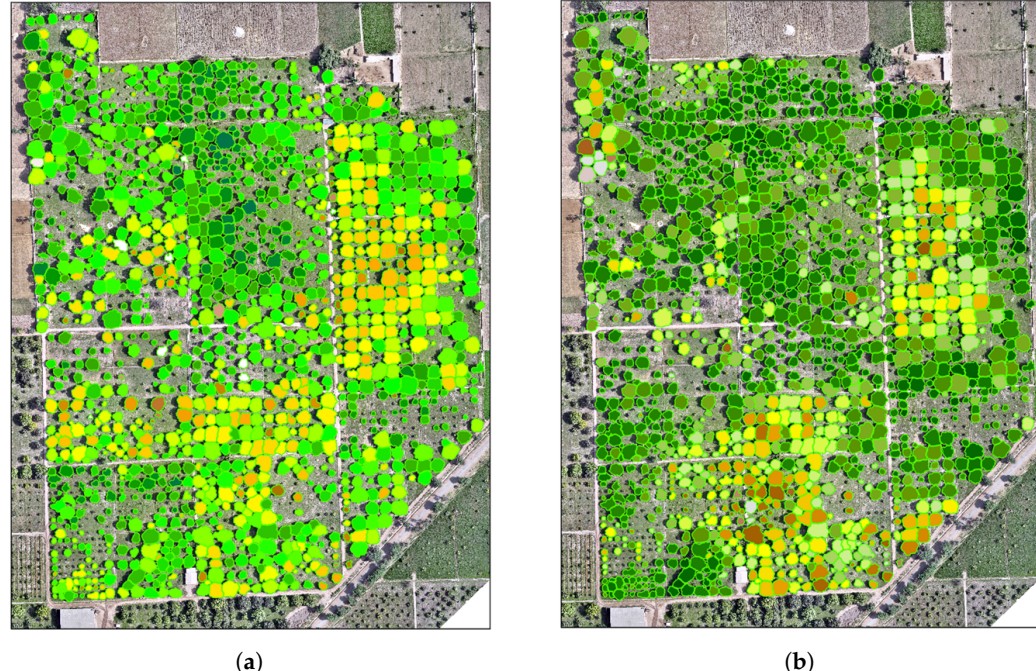

(**a**)          (**b**)

**Figure 10.** Mango tree health estimation based on vegetation indices derived from multispectral UAV imagery and accurate canopy delineation through the proposed framework. (**a**) Mean NDVI (brown to green). (**b**) Mean NDRE (brown to green).

Results indicate that by having accurate delineation of mango tree canopies, the negative impact of surrounding vegetation is mitigated. Thus, the calculation of vegetation indices based on these canopies is not skewed, and more accurate vegetation indices like NDVI and NDRE can be calculated, yielding better indicators of tree health. Hence, better discrimination between the target trees and non-target vegetation is more beneficial for precision agriculture-based orchard management interventions.

These results indicate ways in which the proposed framework can be extended for various diverse applications. Structural and health-related attributes of mango trees can be combined into a holistic analysis for tree-level proactive precision agriculture interventions, even for large orchards, thereby improving yield and economic benefits.

## 4. Conclusions and Future Directions

An automated deep learning scheme for the detection and delineation of mango tree canopies was proposed. The method is tested against a ground truth dataset and closely replicates the manual measurements within acceptable error bounds. The study demonstrates the potential for large-scale zonal analysis. The robustness of the framework is due to the integration of YOLOv7 and SAM with good accuracy and consistency. Details of the chosen hyperparameters are given for reproducing the results in different environmental settings.

Several future directions can be pursued for further refinement. The preprocessing stage may include considerations for lighting conditions, shadows, and image resolution. This could improve segmentation accuracy in varied field conditions. Diversity and size of the dataset can also be increased to include tree canopy variability. This will add more generalizability to the results and ensure that the models perform well across different species and conditions. Outlier analysis and fine-tuning of the model parameters can also be a promising undertaking. The effects of overlapping branches and irregular canopy shapes might be mitigated with error minimization.

Additional preprocessing steps can be introduced to enhance the model's ability to negotiate background noise. Future work may also explore enhancement of speed and accuracy, and benchmarking the pipeline against other state-of-the-art methods for a particular set of downstream tasks. Multi-temporal multi-spectral imagery can also be integrated into the proposed framework for tree growth, health, and yield estimation. The scheme can also evolve into a valuable web-based tool for large-scale ecological and agricultural applications.

**Author Contributions:** Conceptualization, M.M.A.; Data curation, M.S.I.; Investigation, M.M.A.; Methodology, M.M.A. and A.D.B.; Project administration, E.H. and J.I.; Resources, E.H. and J.I.; Software, M.M.A., A.D.B. and M.S.I.; Supervision, E.H. and J.I.; Validation, A.D.B. and M.S.I.; Writing—original draft, M.M.A. and A.D.B.; Writing—review and editing, M.S.I., E.H. and J.I. All authors have read and agreed to the published version of the manuscript.

**Funding:** This research received no external funding.

**Data Availability Statement:** The original contributions presented in the study are included in the article, further inquiries can be directed to the corresponding author.

**Conflicts of Interest:** The authors declare no conflicts of interest.

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
