# Peer review of "High-Precision Mango Orchard Mapping Using a Deep Learning Pipeline Leveraging Object Detection and Segmentation"

_remotesensing, doi:10.3390/rs16173207_

Round 1

Reviewer 1 Report

Comments and Suggestions for Authors

The author attempted to combine the most state-of-the-art AI models including the YOLO7 and SAM model for mango tree detection and crown delineation, which is interesting and practical. However, the innovation and workload seem to be somewhat lacking for the remote sensing journal. This paper draft has a lot of room for improvement, both in terms of experiments and paper organization. The comments and questions are summarized as follows:

The "Introduction" and "Related Work" chapters contain significant overlap in content and ideas. Two chapters can be merged into one or the author can add more introduction about mango tree crown delineation studies. The author broadly discusses the limitations of traditional GIS, remote sensing techniques, and OBIA in tree crown delineation, while also emphasizing the advancements in deep learning, particularly CNNs. However, the author should more effectively summarize the existing literature on mango tree crown delineation (I didn’t see it in the paper draft) and clearly articulate how the proposed model in this study is innovative or distinct from previous work.

Line 30: LiDAR provides precise tree crown lineation results compared to the RGB imagery, here, does the author refer to tree top detection and then applying watershed algorithms?

Line 102: the description here does not seem to be necessary. Or I think the reason why the author used RGB images may be because these two AI models require three-channel import.

Line 250: the author split the entire image into 800*800 tiles. I believe it is due to the limitations of GPU computations. How large is the orthomosaic image? How many tiles were used for model training, validation, and test? The author needs to give a table of the model training and test datasets.

Line 241: in the result and discussion chapter, the author spent a lot of space explaining the definition of accuracy evaluation metrics, which should be added to the method chapter.

As mentioned above, the YOLO7 and SAM process the image of 800*800 pixels. That means, for the entire orthomosaic image, we need to clip it into several tiles, and import each tile to the model and finally merge into a mango tree detection map. How was it implemented in this paper?

Did you the author apply this model to multi-temporal image data to track mango tree growth?

Reviewer 2 Report

Comments and Suggestions for Authors

The authors propose a deep-learning method for accurately delineating tree canopies in mango orchards. It combines object detection with generic semantic segmentation. Object detection relies on YOLOv7 model to detect individual mango trees within UAV imagery and generate bounding boxes around them. Then the Segment Anything Model (SAM) utilizes the detected bounding boxes as prompts to precisely outline the canopy boundaries within those boxes. By combining these two steps, the method aims to overcome challenges associated with traditional methods that suffer from overlapping canopies and complex tree structures. Essentially, the pipeline involves capturing UAV imagery, processing it into an orthomosaic, training the YOLOv7 model to detect trees, using those detections to guide the SAM for precise segmentation, and finally, evaluating the accuracy of the segmented canopies against ground truth data. 

The presented method is valuable and it can certainly drive some practical applications. However, from the scientific point of view, it relies on existing technologies and moreover lacks comparison with other state-of-the-art techniques in the field. For that reason, the proposed manuscript should be submitted for a major revision to address the following issues:

1) Additional information should be provided on how segmentation results can be applied to analyze the state of the orchard. There is a brief description given in section 3.4. that should be extended, and also abstract should include one sentence that will present the goal. You should emphasize the goal of the proposed method. If the goal is to detect trees that are not sufficiently developed, how does the proposed method compare to the other state-of-the-art techniques? The metric for comparison should be picked based on the goal.

2) One of the main issues with the manuscript is that there is no comparison with other state-of-the-art techniques in the field. The authors should provide more details throughout the text on how the proposed method compares to other state-of-the-art techniques in the field. 

3) In the title and throughout the manuscript the term "ensemble deep-learning pipeline" is used. In my opinion, the term "ensemble" is redundant and wrong since it is usually applied to a group of models that work together to improve the results of a single task. However, here we have two different deep-learning models that are used in the pipeline for two different tasks. For that reason, "ensemble" should be removed throughout the text.

4) Figure 5. Subimages are referenced by numbers (1)-(4), but numbers are not shown in the image. Please use letters (a)-(d) instead.

5) Figure 8. Subimages are referenced by letters (a)-(d), but letters are not shown in the image. "Difference FT" is cropped.

Round 2

Reviewer 1 Report

Comments and Suggestions for Authors

The author used 14 image tiles to train the model. I doubt whether this sample size is convincing. Although the Yolo7 and SAM models require less data, the author needs to state the risks behind this sample size. Most other related studies use hundreds or thousands of pictures to make the model stable.

Author Response

Comment 1: The author used 14 image tiles to train the model. I doubt whether this sample size is convincing. Although the Yolo7 and SAM models require less data, the author needs to state the risks behind this sample size. Most other related studies use hundreds or thousands of pictures to make the model stable.

Response 1: Thank you for your insightful suggestion. We converted the image into 16x image tiles, out of those 14x image tiles were used for training purposes and 2x image tiles were used for validation. When training detection models, the sample (trees) should be large enough as you suggested. In line with the observation, these 14 tiles contain over 1500 trees for a single class detector, which gives good accuracy for object detection and segmentation. 

This confusion arose due to a lack of writing clarity on our part and we thank the reviewer for highlighting this. Now it has been rectified and clearly stated in the manuscript as follows:

“…This ensured 87.5% of individual trees (1637) to be included in the training sample and 12.5% (234) trees to be kept for the validation of the output. Detections were performed on each tile measuring 800 × 800 pixels. YOLOv7 with fine-tuned parameters was used to detect a single class of individual mango trees within each tile. The results from all tiles were then appended together to generate a complete detection map for the full orthomosaic image. However, it is important to note that for medium to low-resolution satellite data and/ or detecting more than one object detection/ land use class, a larger number of images/ image tiles will be required to train a YOLOv7 variant.”

 (Reference: Line 163-169)

Reviewer 2 Report

Comments and Suggestions for Authors

The authors addressed all the issues I have listed, so my advice is to accept the paper in its present form.

Author Response

Thank you for your review.